# Assembly of mTORC3 Involves Binding of ETV7 to Two Separate Sequences in the mTOR Kinase Domain

**DOI:** 10.3390/ijms251810042

**Published:** 2024-09-18

**Authors:** Jun Zhan, Frank Harwood, Sara Ten Have, Angus Lamond, Aaron H. Phillips, Richard W. Kriwacki, Priyanka Halder, Monica Cardone, Gerard C. Grosveld

**Affiliations:** 1Department of Genetics, St. Jude Children’s Research Hospital, Memphis, TN 38105, USA; frank.harwood@stjude.org (F.H.); priyanka.halder@stjude.org (P.H.); monica.cardone@stjude.org (M.C.); 2Center for Gene Regulation and Expression, College of Life Sciences, University of Dundee, Dow Street, Dundee DD1 5EH, Scotland, UK; s.m.tenhave@dundee.ac.uk (S.T.H.); a.i.lamond@dundee.ac.uk (A.L.); 3Department of Structural Biology, St. Jude Children’s Research Hospital, Memphis, TN 38105, USA; aaron.phillips@stjude.org (A.H.P.); richard.kriwacki@stjude.org (R.W.K.)

**Keywords:** mTOR, ETV7, mTOC3, rapamycin

## Abstract

mTOR plays a crucial role in cell growth by controlling ribosome biogenesis, metabolism, autophagy, mRNA translation, and cytoskeleton organization. It is a serine/threonine kinase that is part of two distinct extensively described protein complexes, mTORC1 and mTORC2. We have identified a rapamycin-resistant mTOR complex, called mTORC3, which is different from the canonical mTORC1 and mTORC2 complexes in that it does not contain the Raptor, Rictor, or mLST8 mTORC1/2 components. mTORC3 phosphorylates mTORC1 and mTORC2 targets and contains the ETS transcription factor ETV7, which binds to mTOR and is essential for mTORC3 assembly in the cytoplasm. Tumor cells that assemble mTORC3 have a proliferative advantage and become resistant to rapamycin, indicating that inhibiting mTORC3 may have a therapeutic impact on cancer. Here, we investigate which domains or amino acid residues of ETV7 and mTOR are involved in their mutual binding. We found that the mTOR FRB and LBE sequences in the kinase domain interact with the pointed (PNT) and ETS domains of ETV7, respectively. We also found that forced expression of the mTOR FRB domain in the mTORC3-expressing, rapamycin-resistant cell line Karpas-299 out-competes mTOR for ETV7 binding and renders these cells rapamycin-sensitive in vivo. Our data provide useful information for the development of molecules that prevent the assembly of mTORC3, which may have therapeutic value in the treatment of mTORC3-positive cancer.

## 1. Introduction

mTOR plays a crucial role in cell growth by regulating various processes, including ribosome biogenesis, metabolism, autophagy, mRNA translation, and cytoskeleton organization [1,2,3]. It is a 289 kDa serine/threonine kinase that is part of two distinct, extensively described protein complexes, mTORC1 and mTORC2. mTOR belongs to the phosphatidylinositol 3-kinase (PI3K)-related kinase (PIKKs) family, along with ATM, ATR, DNA-PK, SMG1, MEC1, RAD3, MEI41, and TRRAP (a pseudokinase), which share a catalytic domain of the phosphatidylinositol 3′ (lipid) kinases [4,5,6].

The 1345 amino terminal residues of mTOR contain 32 HEAT repeats, each representing a 30- to 40-residue segment consisting of two antiparallel alpha helices connected by a flexible linker [7,8]. The HEAT domains can be subdivided into the N- and M-HEAT domains [9], serving as protein–protein interaction domains [6]. This is succeeded by the roughly 500-residue FAT (FRAP/ATM/TRRAP) domain, followed by the mTOR catalytic domain. The mTORC1 cryo-EM structure shows that the catalytic domain has a canonical two-lobed structure with key insertions in both the amino-terminal and the carboxy-terminal lobes [10]. The N-terminal lobe of the catalytic domain contains the FKBP-12-rapamycin-binding domain (FRB), while the C-terminal lobe contains the LBE element, which binds mLST8.

The FRB domain extends over the catalytic cleft of the amino terminal lobe, making the cleft deeply recessed. Modeling of the FKBP12:rapamycin complex onto the FRB segment indicates that the latter severely limits access to the catalytic cleft. Moreover, the side of the FRB domain facing the catalytic cleft serves as a secondary substrate-binding site [11,12]. Recently, we identified a rapamycin-resistant mTOR complex, called mTORC3, which assembles in the cytoplasmic fraction of ETV7-expressing cells [13].

ETV7 is a member of the ETS (E26-transforming-specific) family of transcriptional regulators [14], which have collectively been implicated in a wide range of developmental and disease processes. The ETV7 gene is located within the MHC cluster at 6p21 [15,16]. It consists of nine coding exons that are alternatively spliced [16,17,18]. Splicing together exons 1–8 produces the most abundant transcript, while skipping exon 8 produces a transcript of minor abundance containing exon 9. The key functional domains of ETV7 include an N-terminal pointed (PNT) domain, a much less-conserved central domain, followed by a conserved ETS domain and a smaller C-terminal inhibitory domain (CID) that is missing in the product encoded by the exon 9 transcript [15,16,19,20].

The PNT domain of ETV7 consist of four helices that are responsible for homodimerization [16,21], as well as heterodimerization with ETV6 [15,16]. The ETS domain [15] has a winged helix–turn–helix topology that facilitates binding to promoter or enhancer elements of target genes. ETV7 binds strongly to a core -ccGGAAgt- sequence, but it can also interact with a variety of related sequences [15,22]. Although the PNT and ETS domains are separated by 106 amino acids in the linear sequence, the prediction of the AlphaFold 3D structure of ETV7 suggests that these domains may transiently experience electrostatic interactions, possibly involving R48 in the PNT domain and D222 in the ETS domain.

Cells expressing ETV7 assemble mTORC3, whose kinase activity induces increased proliferation and renders cells resistant to rapamycin, indicating that inhibiting mTORC3 may be of therapeutic value for the treatment of cancer. In this manuscript, we describe a more precise identification of the sequences in ETV7 and mTOR that mediate their interaction. Using mutant mTOR and ETV7 in in vivo and in vitro experiments, we show that the FRB and LBE domains in the mTOR kinase domain interact with the PNT and ETS domains of ETV7, respectively. In addition, we show that overexpression of a C-terminal extended FRB domain alone interferes with mTORC3 assembly and renders rapamycin-resistant cells rapamycin-sensitive. These insights provide a first step on the path to designing mTORC3-specific inhibitors.

## 2. Results

### 2.1. The PNT and ETS Domains Are Important for mTORC3 Assembly

Previously, we reported that deletions in the PNT domain of ETV7 of three and nine amino acids, respectively, blocked the assembly of mTORC3 [13]. This supported the direct involvement of the PNT domain in mTOR binding. We also reported mutations in the ETS domain of ETV7 that abolished its DNA binding activity but cells expressing this mutant assembled mTORC3, suggesting that mTORC3 assembly is independent of ETV7′s transcriptional activity [13]. However, an ETV7 deletion mutant containing only the PNT domain (ETV7^M1-E144^) failed to generate mTORC3 activity because rapamycin-sensitive KE7^-^ cells (a Karpas-299 derivative that did not express ETV7, [13]) expressing this mutant remained sensitive to rapamycin (Figure 1A, left panel). This was confirmed by a negative reciprocal co-IP of FLAG-tagged ETV7^M1-E144^ and mTOR (Figure 1A, right panel). Given the poor conservation of the amino acid sequence linking the PNT and ETS domains of ETV7 in seven vertebrates (19.8% similarity, and 5.7% identity between human, chimpanzee, dog, squirrel, Tasmanian devil, chicken, and zebrafish) [23] (Appendix A), this result opened the possibility that the ETS domain also contributes to mTORC3 assembly. The PNT domain has more than 70% similarity and the ETS domain has more than 80% similarity among these seven vertebrates (Appendix A). Given this high level of conservation, we tested whether FLAG-tagged zebra fish ETV7 (zETV7) assembled mTORC3 in KE7^-^ cells. The similarity of human and zebrafish ETV7 is 81% in the PNT and 91% in the ETS domain, but only 32% in the intervening sequence (Appendix A). We selected a single-cell clone expressing moderate levels of zETV7, called KE7^-^-zETV7, and performed rapamycin response experiments [13] comparing KE7^-^ and KE7^-^-zETV7, with Karpas-299 cells. As reported [13] and shown in Figure 1B, KE7^-^ was sensitive and Karpas-299 resistant to rapamycin, while KE7^-^-zETV7 cells showed intermediate resistance. This indicated that zETV7 assembled mTORC3 in human cells. In fact, reciprocal co-immunoprecipitation showed that FLAG-zETV7 pulled down human mTOR and vice versa (Figure 1B, right panel). The involvement of the PNT and ETS domains of ETV7 in mTOR binding was further supported by testing the resistance to rapamycin of KE7^-^ cells expressing alternatively spliced ETV7 in which the predominantly used last exon of *ETV7* (exon 8) was replaced by the less frequently used exon 9 (Appendix A), which encodes a nonhomologous C-terminus. Despite the presence of ETV7 exon 9 transcripts in humans, the evolutionary conservation of this exon is low even among primates and between primates and monkeys (Appendix A) and a splice form-specific function of this variant remains to be determined. The rapamycin sensitivity experiment shows that the resistance to rapamycin of ex9-ETV7 mTORC3 is like that of ex8-ETV7 mTORC3 (Figure 1C). Reciprocal co-immunoprecipitation experiments further confirmed the ex9-ETV7-mTOR interaction (Figure 1C, right panel). Together, this suggests that the exon 8 or 9 encoded C-terminus of ETV7 does not contribute to mTORC3 assembly.

### 2.2. Point Mutations in the PNT and ETS Domains of ETV7 Identify Amino Acids Involved in the Assembly of mTORC3

Given that the PNT and ETS domains of ETV7 bind to mTOR, we wished to identify amino acid residues in these domains that are involved in mTORC3 assembly. First, we generated alanine substitutions in the PNT domain of ETV7. The PNT domain mediates protein–protein interactions and contains two interaction surfaces, ML and EH (Figure 1D, highlighted in red and yellow, respectively) [24]. These surfaces are involved in the formation of ETV7 dimers or oligomers/multimers in vitro, depending on the concentration of the protein. We generated four mutants (M82A, I89A, V105A, L109A), two in the ML and two in the EH surface (Figure 1D). We introduced individual mutants into KE7^-^ cells and tested selected clones for resistance to rapamycin (Figure 1D). Rapamycin resistance indicates assembly of mTORC3, while rapamycin sensitivity implies no assembly. All four mutants were sensitive to rapamycin, thereby precluding mTORC3 assembly, indicating that these amino acid residues in both the ML and EH surfaces contribute to the assembly of mTORC3 (Figure 1D). The reciprocal co-IP experiments corroborated the rapamycin resistance data by showing that none of the mutants coprecipitated with mTOR and vice versa (Figure 1D, right panel). Next, we investigated which amino acids in the ETS domain participate in the assembly of mTORC3. We applied homology modeling using SWISS-MODEL [25] with the 3D structure of the ETV6 ETS domain bound to DNA [26] as a template. This identified nine hydrophobic amino acids residues in the ETV7 ETS domain (L235/Y231/Y239/Y242/L264/W227/L289/I291/L305) that are solvent-exposed and cluster in three hydrophobic patches (Figure 2A). To avoid an energetically unfavorable state, these amino acids are likely to engage in protein–protein interactions and are potentially involved in mTOR binding. We generated single alanine substitutions for each of the nine amino acids (Figure 2B), introduced each mutant into KE7^-^ cells, and selected single-cell clones expressing these mutants. Only Y239A and L289A were rapamycin-resistant, thus excluding a role in mTORC3 assembly. The other seven (L235/Y231/Y242/L264/W227/I291/L305) were sensitive to rapamycin, suggesting that these amino acids participate in mTORC3 binding (Figure 2C). This was confirmed by immunoblotting reciprocal co-IP experiments, which showed that rapamycin-resistant mutants co-IP ETV7 and mTOR, whereas non-resistant mutants did not (Figure 2C). These results are unlikely to be caused by overt changes in subcellular localization of the different ETV7 mutants, as all of them showed both nuclear and cytoplasmic localization like wild-type ETV7 (Appendix A) and mutations to alanine at all positions were not predicted to significantly destabilize the protein fold based on calculations using FoldX [27] (Appendix A). Taken together, we have identified amino acid residues in both the PNT and ETS domains involved in mTORC3 assembly.

### 2.3. The FRB Domain in the N-Terminal Kinase Lobe of mTOR Interacts with ETV7

Next, we investigated which mTOR sequences are involved in binding to ETV7. Given that ETV7 counteracts mTOR’s kinase sensitivity to rapamycin, we hypothesized that ETV7 might bind the mTOR kinase domain. A FLAG-tagged expression construct for the mTOR kinase domain (amino acids 1363–2549) was introduced into Karpas-299 and the immunoprecipitated FLAG immunoblotted material was probed with FLAG and ETV7 antibodies. This confirmed that ETV7 co-immunoprecipitated with the mTOR kinase domain (Figure 3A). To map which sequences in the mTOR kinase domain bind ETV7, we first confirmed that the in vivo association could be recapitulated in vitro. We co-incubated the recombinant mTOR kinase domain (amino acid 1363-2549) and the ETV7 protein and investigated the formation of the complex by Western blot analysis of protein immunoprecipitated with the ETV7 antibody. This showed that ETV7 IPs pulled down mTOR in a dose-dependent and saturable manner (Appendix A). We then used this assay to map the mTOR sequences involved in ETV7 binding.

Given that rapamycin resistance of TORC2 in yeast is caused by blocking access of FKBP12–rapamycin to the FKBP–rapamycin binding domain (FRB) in the N-terminal lobe of the TOR kinase domain by Avo3 (yeast Rictor) [28], we postulated that ETV7 might similarly block binding of FKBP12–rapamycin to the mTOR FRB domain in mTORC3. If so, adding an increasing amount of FKBP–rapamycin to the ETV7–mTOR association reaction would progressively inhibit ETV7–mTOR binding. Indeed, a 10-fold FKBP–rapamycin molar excess in the reaction reduced ETV7–mTOR association, while a 100-fold FKBP–rapamycin molar excess prevented interaction (Figure 3B), suggesting the involvement of the FRB region in mTOR–ETV7 binding. Next, we tested whether the FRB fragment alone could associate with ETV7. We showed (Appendix A) that a FLAG-tagged in vitro translated FRB fragment bound to ETV7. However, the avidity of association was not robust, causing variable co-IP results between experiments, suggesting that additional sequences flanking the FRB domain might also contribute to ETV7 binding. We tested this by repeating the experiment with an FRB fragment of 283 amino acids carrying a 90 amino acid N-terminal extension and a 96 amino acid C-terminal extension, called FRB-new (Figure 3C), which consistently associated with ETV7. Subsequently, we generated different deletions of the FRB-new fragment and assessed the in vitro association with ETV7. We found that a 90 amino acid N-terminal truncation of the FRB-new fragment, called Ndel-FRB, showed the strongest association with ETV7 (Figure 3C). To find the smallest fragment that still could associate with ETV7, we made serial deletions from both the N- and C-terminal ends of Ndel-FRB. This identified a minimal binding fragment of 105 amino acids (Figure 3D), consisting of the 70 C-terminal amino acids of FRB C-terminally extended with 35 amino acids non-FRB amino acids.

### 2.4. The mLST8 Binding Element (LBE) of mTOR Also Associates with ETV7

Unlike mTORC1 and mTORC2, mTORC3 does not contain mLST8 [13], an obligate activating subunit of mTORC1/2 [29,30]. mLST8 binds to the 39 amino acid LBE domain in the C-terminal lobe of the mTOR kinase domain. We tested whether ETV7 replaces mLST8 in mTORC3 by binding to LBE sequences. Therefore, we generated an N-terminal (37 amino acids) and C-terminal (44 amino acids) extended LBE protein, totaling 120 amino acids, and showed that it is associated with ETV7 in vitro (Figure 4A). As a control, we used a fragment of RUVBL2 that does not interact with ETV7. Testing the association of subsequent N- and C-terminal truncations of this 120 amino acids LBE fragment showed that the 6 central amino acids of LBE (DHLTLM) are essential and sufficient to bind ETV7 (Figure 4B). Next, we tested whether FRB and LBE can bind ETV7 simultaneously. Co-incubation of ETV7 with the Ndel-FRB and extended LBE fragments demonstrated that both associated with ETV7 (Figure 4C); again, the RUVBL2 fragment served as a control. To address whether the FRB and LBE binding sites in ETV7 overlap or not, we repeated this experiment by keeping the amount of LBE constant but increasing the amount of FRB five times and vice versa. This showed that FRB and LBE did not compete for ETV7 binding (Figure 4C), supporting the notion that FRB and LBE engage in nonoverlapping amino acid sequences in ETV7. To delineate which ETV7 sequences FRB and LBE bind, we used several ETV7 fragments, ETV7ex1-4, ETV7PNT, and ETV7ex5-8 (Appendix A). This showed that FRB only bound to ETV7 fragments that contained the PNT domain (Figure 4D), while LBE was associated with fragments that contain either domain, but it associated slightly better with fragments that contained the ETS domain, while the control RUVBL2 fragment did not associate with ETV7.

### 2.5. Cross-Link Mass Spectrometry Identifies Amino Acid Residues Involved in ETV7 mTOR Association

To better understand how ETV7 and mTOR interact within the mTORC3 complex, information important for the future design of small molecules that would disrupt the complex or prevent its assembly, we determined how ETV7–FRB contact each other at the amino acid level. To this end, we performed mass spectrometry identification of peptides derived from the cross-linked ETV7–FRB-new complex. Since both the ETV7 PNT domain and the mTOR FRB fragment harbor relatively few lysine, while arginine residues are more abundant, we used the 4,4’-diglyoxyloylbiphenyl arginine crosslinker (Appendix A) [31]. The optimization of the cross-linking conditions is shown in Figure 5A. After cross-linking, the sample was subjected to PAA electrophoresis, stained by Coomassie Brilliant Blue, and the gel slice containing the presumptive cross-linked dimers was cut from the gel (Figure 5B). Subsequently, samples were processed by mass spectrometry. The cross-link mass spectrometry data are summarized in Table 1. In ETV7, the cross-linked sites were mainly located in the PNT domain (Figure 5C, Appendix A), which confirmed its specificity for binding the mTOR FRB domain. In FRB, the cross-linked sites are within the FRB sequence itself, but also in the C-terminal extension (Figure 5C, Appendix A), showing that its arginine residues are spatially close enough to arginine residues in the ETV7 PNT domain to be cross-linked. Furthermore, these data are consistent with our in vitro IP association data.

### 2.6. Overexpressed Ndel-FRB Domain Outcompetes mTOR for ETV7 Binding and Renders Karpas-299 Cells Rapamycin-Sensitive

Given that FRB binds ETV7, we tested whether FRB could compete with mTOR for ETV7 binding and, thus, disrupt mTORC3 activity. We first examined the mTOR–ETV7 association in vitro in the presence of Ndel-FRB and found that mTOR binding of mTOR to ETV7 was reduced in the presence of Ndel-FRB (Figure 6A), suggesting that FRB competes with mTOR for binding to ETV7 in vitro and could potentially reduce mTORC3 assembly in vivo. Next, we tested whether Ndel-FRB expression in vivo could affect mTORC3 activity and, make these cells sensitive to rapamycin. We introduced a FLAG–Ndel-FRB (192 amino acids) expressing lentiviral vector (Appendix A) in Karpas-299 and KE7^-^ cells and selected two pools of resistant cells of each, one pool transduced once (1 × FRB) and the other transduced twice (2 × FRB). We treated Karpas-299 parental cells, and each of the pools with an increasing concentration of rapamycin, and found that, like KE7^-^ and K-E7^-^–Ndel-FRB, the pools of Karpas-299–Ndel-FRB cells had become rapamycin-sensitive, as they stopped proliferating at a concentration of 1ng/mL rapamycin (Figure 6B). This indicated disruption of mTORC3, and immunoblots of Karpas-299, K-E7^-^, 1 × FRB (Ndel-FRB) Karpas-299, and 1 × FRB K-E7^-^ probed for p-p70S6K and total p70S6K, showed that like KE7^-^ and K-E7^-^–Ndel-FRB, Karpas-299–Ndel-FRB but not Karpas-299 showed loss of phosphorylation of p70S6K–Thr389 at rapamycin concentrations > 3 ng/mL (Figure 6C; [13]). Reciprocal co-IPs of mTOR and ETV7 confirmed that the Ndel-FRB fragment interfered with mTORC3 assembly in Karpas-299–Ndel-FRB cells, as ETV7 and mTOR did not co-immunoprecipitate (Figure 6D). To support these data, we also expressed the FRB deletion mutant ND72C61 in Karpas-299 cells, which no longer binds to ETV7 in vitro (Figure 3D) and, therefore, should not interfere with the assembly of mTORC3 in vivo. In fact, cells expressing the ND72C61 fragment were insensitive to rapamycin inhibition (Figure 6B) and maintained p70S6K–Thr389 phosphorylation (Figure 6C), showing that the ND72C61 fragment cannot interfere with the assembly of mTORC3. Furthermore, reciprocal mTOR and ETV7 co-IP experiments showed maintenance of co-IP of mTOR and ETV7 in Karpas-299–ND72C6 cells (Figure 6D). To ensure that the Ndel-FRB and ND72C61 fragments were expressed at similar levels, lysates of equal numbers of KE7^-^–Ndel-FRB, Karpas-299–Ndel-FRB, and Karpas-299–ND72C61 cells were immunoblotted, and using an anti-FLAG antibody, equal expression was confirmed (Figure 6E). Together, this suggests that the Ndel-FRB fragment, its derivatives, or a small molecule mimicry of this fragment, could inhibit mTORC3 and could be of potential therapeutic value.

## 3. Discussion

Previously, we showed that the cytoplasmic fraction of the ETV7 transcription factor associates with mTOR and assembles an alternative mTOR complex, mTORC3, that stimulates cell growth, induces resistance to rapamycin, and accelerates tumorigenesis [13]. In the nucleus, ETV7 mostly functions as a transcriptional repressor and is known to be upregulated in various types of cancer [32]. Its expression also correlates with increased tumor aggressiveness [33], affecting various molecular and cellular pathways [32,34]. Given the role of mTORC3 in tumorigenesis, we embarked on an effort to define which sequences in mTOR and ETV7 mediate their mutual binding, with the goal of identifying ways to interfere with this association and prevent the assembly of mTORC3.

As proof for the presence of mTORC3, we test for rapamycin resistance and reciprocal co-IP of ETV7 and mTOR. We believe that these tests are robust, as we never found an example of rapamycin resistance without reciprocal co-IP or vice versa. As mentioned in the introduction, ETV7 consists of a PNT, a linker, and an ETS domain. Our previous research had indicated that deletions within the ML surface of the PNT domain prevented the assembly of mTORC3. Because these mutations were coarse and perhaps altered the 3D structure of the domain, we tested whether more subtle single alanine substitution mutations would similarly affect the assembly of mTORC3. The PNT domain mediates protein–protein interactions and contains two interaction surfaces, ML and EH [23]. By introducing four-point mutations in the PNT domain, two on the ML surface (M82A, I89A) and two on the EH surface (V105A, L109A), we were able to disrupt the assembly of mTORC3, suggesting that both surfaces are involved in the association with mTOR. How exactly these surfaces bind mTOR needs to be determined via 3D protein structure analysis, which is beyond the scope of the current work.

Initial work using the ETV7 PNT fragment (ETV7^M1-E144^), which failed to assemble mTORC3, prompted us to determine whether the ETS domain contributed to the assembly of mTORC3. Computational modeling helped us identify nine hydrophobic amino acid residues in the ETV7 ETS domain (L235/Y231/Y239/Y242/L264/W227/L289/I291/L305) that are solvent-exposed and cluster in three hydrophobic patches. Our data reveal that only the Y239A and L289A mutations confer rapamycin resistance, while the other seven (L235/Y231/Y242/L264/W227/I291/L305) remain sensitive to rapamycin and disrupt the ETV7–mTOR association, suggesting their involvement in mTORC3 binding. Y293 appears to be more buried in the structure (Figure 2A) with, according to the AlphaFold model, its tyrosyl side chain pointing inward, which would make it unavailable for interacting with LBE sequences; hence, the absence of an effect by its change to alanine. Why the L289A mutation has no effect awaits the availability of a 3D structure, just as the structure may explain the effects of the other seven hydrophobic amino acid changes. Utilizing cross-linking followed by mass spectrometry technology, we mapped arginine residues in the PNT domain that are close enough to the mTOR FRB domain to be cross-linked and the data closely align with our association data. This highlighted the importance of the PNT domain of ETV7 in the association reaction. Although there was no overlap between the cross-linking data and the point mutations in the PNT domain, this discrepancy may be attributed to the use of an arginine cross-linking reagent, which predominantly identifies arginine residues. Nevertheless, the information will be valuable for later therapeutics design.

Although we currently do not possess a model that illustrates how ETV7 and mTOR interact with each other, the structure of mTOR is known [10]. Cryo-electron microscopy studies of mTORC1 have revealed that its catalytic domain exhibits a canonical two-lobe structure containing key insertions in the amino-terminal and carboxy-terminal lobes [10]. The FRB domain in the N-terminal lobe extends over the catalytic cleft, along with the FATC and LBE domains extending from the carboxy-terminal lobe. Binding ETV7 to the FRB domain would compete with FKBP12–rapamycin to binding to the FRB domain. This possibility is supported by the fact that increasing the amount of FKBP12–rapamycin in an in vitro association reaction successfully prevented ETV7 from binding to the mTOR kinase domain (Figure 3B). Additionally, cross-linking experiments have further confirmed the importance of the FRB domain, including that of its downstream 90 amino acid residues for its association with ETV7.

Across from the FRB domain in the 3D structure of mTOR lies the LBE domain, which interacts with mLST8. mLST8 is a necessary activating subunit of mTOR complexes [29,30]. The structure suggests that the extended interaction surface of mLST8 may directly stabilize the LBE structure and indirectly influence the organization of the mTOR active site through the LBE–FATC–catalytic spine of interactions. Since mTORC3 lacks mLST8, we hypothesized that ETV7 competitively binds to the LBE domain of mTOR. Our in vitro data support this hypothesis. However, the interaction seems weaker than the PNT–FRB binding and the LBE fragment appears sticky in our co-IP experiments, as its co-IP signal with ETV7 fragments containing the ETS domain is only marginally stronger than that with those not containing the ETS domain (Figure 4D). We propose that the ETS domain of ETV7 mimics the function of mLST8 by binding to the LBE sequence. Provided our LBE interaction data are correct, it would open the possibility that a single ETV7 molecule could straddle the mTOR kinase cleft and bind to the FRB and LBE sequences, simultaneously.

In our in vitro association experiments, we identified a fragment called Ndel-FRB, which showed the strongest binding to ETV7 in vitro. Testing of this fragment for its ability to disrupt mTORC3 assembly in Karpas-299 cells indeed showed that the cells slowed proliferation, had become sensitive to rapamycin, and that ETV7 and mTOR no longer co-immunoprecipitated (Figure 6), while a ND72C61 deletion mutant that does not bind to ETV7 in vitro, could not disrupt mTORC3 assembly in vivo. A caveat with these experiments is that, despite many attempts, we were unable to show co-IP of Ndel-FRB with ETV7. We believe that this may be due to the ETV7–Ndel-FRB complex masking the epitopes for immunoprecipitation.

We are well aware of the fact that in vitro interactions between fragments of proteins show their potential to interact, but 3D modeling, or a 3D structure are needed to show if these interactions remain valid in the actual mTOR3 complex. It should also be noted that this work only describes sequences of ETV7 and mTOR binding to each other, but in vivo, this binding occurs in the context of mTORC3 which, given its estimated size of 1.4 MDa [13], contains other unknown proteins that can affect interactions between ETV7 and mTOR. In addition, we only showed the inhibitory activity of Ndel-FRB in Karpas-299 cells, but further experiments need to be performed in other cell lines and primary tumor cells to determine whether our observations are applicable to mTORC3 expressing cells in general.

Based on our discoveries, we believe it would be valuable to design small peptides that can penetrate cell membranes for the therapeutics of cancers expressing mTORC3. In addition to the design of peptide-based therapeutics, we propose using time-resolved fluorescence resonance energy transfer (TR-FRET) [35] to screen small molecules that can disrupt the assembly of mTORC3. By establishing a suitable assay system, we can screen compound libraries with the aim of discovering novel small molecules for cancer therapy.

In conclusion, we have successfully mapped domains and amino acid residues involved in the association between ETV7 and mTOR. This information provides valuable insights for the design of therapeutic strategies targeting cancers that express mTORC3.

## 4. Materials and Methods

### 4.1. Antibodies

The following antibodies were used: anti-mTOR, anti-FRAP (Santa Cruz Biotechnology, Dallas, TX, USA), anti-phospho-P70S6KThr389, (Cell Signaling, Danvers, MA, USA), anti-FLAG-M2, and anti-ETV7 (Sigma, Cat #14793, Sigma-Aldrich, Burlington, MA, USA). The rat monoclonal anti-ETV7 peptide antibody (7E4) was previously reported [13]. Detection of the 7E4 immunoprecipitated ETV7 protein was performed on most immunoblots with the Sigma ETV7 antibody (#HPA 029033).

### 4.2. Recombinant Proteins

mTOR kinase domain protein was purchased from Sigma-Aldrich (Cat #14-770, St. Louis, MO, USA). ETV7 protein (Cat #TP307742) and RUVBL2 (Cat #TP300933) were purchased from Origene (Rockville, MD, USA).

### 4.3. Plasmid Constructs

The ETV7 mutations were constructed using the pCL20 vector backbone with ETV7 driven by the TK promoter and the blasticidin selectable marker driven by the mouse actin promoter [13]. All point mutations were generated using the QuickChange lightning site-directed mutagenesis kit (Agilent, Santa Clara, CA, USA). For protein expression in *E. coli,* we used the pET15 b vector. All DNA fragments encoding different proteins or peptides were purchased from integrated DNA technologies (IDT, Research Triangle Park, NC, USA) and inserted into pET15b at the BamHI site using Gibson cloning. The Ndel-FRB fragment followed by a P2A peptide and a puromycin selectable marker (purchased from IDT) was inserted at the AarI/EcoRI sites of the episomal vector EBNA-px330, containing the Epstein–Barr virus *EBNA* gene and the oriP origin of replication [13], producing px330 N-del FRB-Blast-EBNA. The XbaI-EcoRI fragment from this vector, containing the CMV enhancer, the chicken beta-actin promoter, the Ndel-FRB fragment, the p2A sequence, and the puromycin gene were inserted at the NheI-EcoRI sites of Lenti-EFalpha-VP64-dCas9-VP64_Blast (Addgene, #192650, Watertown, MA, USA), which produced LentiFRB-new-puro (Appendix A). The ND72C61 fragment was obtained from IDT and cloned into the EcoRI site of pCl20–Tkp–EcoRI–Actinp–blasticidin. The 3′ terminal half of the mTOR cDNA, starting at the internal EcoRI site, fused at the 5′ end to a FLAG oligo nucleotide, was purchased from IDT and inserted at the EcoRI site pCl20c lentiviral vector, yielding plasmid pCL20-mTOR KD-FLAG. The ETV7^M1-E144^ DNA fragment and the zETV7 cDNA were purchased from IDT and inserted at the EcoRI site of pCL20c–blasticidin lentiviral vector.

### 4.4. Computational Modeling

A homology model of the ETS domain of ETV7 was created based on the available ETV6 structure using SWISS-MODEL [10] with the DNA modeled based on the complex structure of the ETV6 ETS domain. The structure was searched for surface exposed hydrophobic or aromatic residues that clustered in patches away from the DNA binding site. Three such patches were identified consisting of (1) Y239/Y242/L264, (2) Y231/L235, and (3) W227/L289/I291/L30, each of which could represent protein–protein interaction sites. Calculations using FoldX based on the Alpha Fold model of the ETS domain (Q9Y603, residues 223-305 [36]) predicted that each of these alanine mutations are not likely to significantly destabilize the protein structure. Notably, mutations with very similar ΔΔG values, Y239 and Y242, have disparate functional outcomes.

### 4.5. Cross-Linking

ETV7 (500 ng) and FRB-new (400 ng) proteins were co-incubated in CHAPS buffer [40 mM Hepes (pH 7.4), 1 mM EDTA, 120 mM NaCl, 10 mM sodium pyrophosphate, 10 mM β-glycerophosphate, 0.3% CHAPS, 50 mM NaF, 1.5 mM NaVO, 1 mM PMSF] at 40 °C overnight. For arginine cross-linking, 500 µM 4,4’ diglyoxyloylbiphenyl was added and incubated at 30 °C for 2 h. The cross-linked sample was subjected to SDS-PAGE and after Coomassie Brilliant Blue staining the predicted bands were cut out from the gel for mass spec analysis.

### 4.6. Mass Spec Analysis of Cross-Linked Proteins

The gel samples were digested in gel using trypsin after reduction and alkylation (iodoacetamide and dithiothreitol). The resulting samples were analyzed using a high accuracy method (high accuracy MS and MS/MS). The raw files were processed using MaxQuant to obtain a mass list; this was then analyzed in Xi [37] and checked against a database consisting of all 8 isoforms of ETV7 and mTOR.

### 4.7. Protein Purification

A colony of BL21(DE3) carrying a pET15b vector encoding recombinant protein was selected and cultured overnight in Luria broth (LB) containing 100 ug/mL of ampicillin. The next morning, a 1:50 dilution was transferred to 2-L LB, 50 ug/mL ampicillin and incubated in an orbital shaker at 250 RPM, 37 °C. When the cell density reached an OD_600_ of 0.8–1, 1 mM IPTG was added to induce protein expression at 30 °C, shaking at 200 RPM for 2 h. The cells were pelleted, resuspended in 8 mL of pre-cooled lysis buffer (20 mM Tris pH8.0, 150 mM NaCl, 30 mM Imidazole), and supplemented with EDTA-free protease Inhibitor Cocktail (Thermo Fisher Scientific, Waltham, MA, USA). The samples were subjected to 6 rounds of 10 second sonication each at power 3 in a 550 Sonic Dismembrator (Fisher Scientific, Hampton, New Hampshire, USA). Between each round of sonication, the samples were cooled for 1 min on ice. Triton-X-100 was added to the samples to a final concentration of 1%, followed by incubation on ice for 30 min to solubilize the protein. The samples were cleared by a 30-min centrifugation step at 20,000× *g*, 4 °C. Two mL of a 50% slurry of washed Ni-NTA beads (Qiagen #30210, Hilden, Germany) was added to the supernatant of the 2-L culture, followed by an overnight incubation in a rotator mixer at 4 °C. The beads were washed three times with 16 mL of lysis buffer, supplemented with 1% Tritin-X-100. The protein was recovered from the beads using three rounds of elution with 1 mL buffer (20 mM Tris pH8.0, 150 mM NaCl, 250 mM Imidazole). Finally, the protein was dialyzed in 0.5× PBS using a Slide-A-Lyzer Dialysis Cassette (Thermo Fisher Scientific, Waltham, MA, USA).

### 4.8. Cell Culture

Karpas-299, K-E7^-^, and K-E7^-^–ETV7 (13) cells were cultured in RPMI-1640 medium (Corning, New York, NY, USA) supplemented with 10% bovine calf serum (BCS) (Thermo Fisher Scientific), 50 mM GlutaMAX (Life Technologies, Carlsbad, CA, USA), penicillin (100 U/mL), and streptomycin (100 µg/mL) (Life Technologies) in a humidified incubator at 37 °C, 5% CO_2_.

### 4.9. Rapamycin Resistance Assay

For lentivirus production, 293T cells were transiently transfected with pCL20c lentiviral vectors encoding different mutant ETV7 constructs [13]. Subsequently, KE7^-^ cells were transduced with each of the pCL20c–ETV7–Actp–blasticidin lentiviral vectors and selected for 1 week with 1 µg/mL blasticidin (Fisher Scientific). Resistant cells were seeded in methylcellulose [1.5% methylcellulose (Sigma-Aldrich, M0512, RPMI, 10% FBS, 1 µM thymidine and 30 µM uridine], and 2 weeks later, 10 single clones were selected, expanded, and tested for ETV7 expression on Western blot. We selected clones with an ETV7 expression level closest to that of Karpas-299 and used them for rapamycin sensitivity assays as described [13]. Rapamycin resistance experiments using an increasing dose of rapamycin (0, 0.1, 0.3, 1, 3, 10, 100, and 1000 ng/mL) were conducted as described previously [13]. Karpas-299 cells were transduced with Lenti-FRB-new-puro lentivirus (Appendix A) and selected for 2 weeks with 1 µg/mL puromycin. The pool of selected cells was then used for a rapamycin resistance experiment as described above.

### 4.10. Western Blot and Co-IP

Western blot and co-IP analysis were performed as described [13].

## Figures and Tables

**Figure 1 ijms-25-10042-f001:**
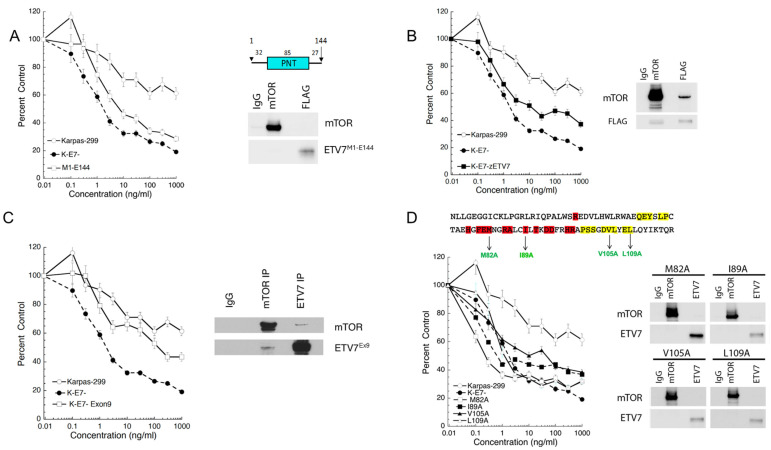
The PNT domain of ETV7 contributes to the assembly of mTORC3. (**A**). Left panel; rapamycin response curve of Karpas-299, KE7^-^, and KE7^-^-ETV7^M1-E144^ cells treated with an escalating dose of rapamycin (0.1, 0.3, 1.3, 10, 30, 100, 300, and 1000 ng/mL) for three days or three population doublings. Cell densities (percent control) as the percentage of cells treated with vehicle. Data are means ± SEM from three independent experiments. Top right; schematic representation of the flag-tagged ETV7^M1-E144^ fragment with numbers indicating the number of amino acids in the different segments of the protein. Underneath the drawing is the immunoblot of the lysate of KE7^-^-ETV7^M1-E144^ cells immunoprecipitated with anti-mTOR or anti-FLAG antibodies and non-relevant IgG, probed for mTOR and FLAG. (**B**). Left panel: rapamycin response curves of Karpas-299, KE7^-^, and KE7^-^-zETV7 cells treated with an increasing dose of rapamycin (0.1, 0.3, 1.3, 10, 30, 100, 300, and 1000 ng/mL) for three days or three population doublings. Cell densities (percent control) as the percentage of cells treated with vehicle. Data are means ± SEM from three independent experiments. Right panel: immunoblot of KE7^-^-zETV7 cell lysate immunoprecipitated with anti-mTOR or anti-FLAG antibodies and non-relevant IgG, probed for mTOR and FLAG. (**C**). Left panel; rapamycin response curves of Karpas-299, KE7^-^, and KE7^-^-ETV7^Ex9^ cells treated with an escalating dose of rapamycin (0.1, 0.3, 1.3, 10, 30, 100, 300, and 1000 ng/mL) for three days or three population doublings. Cell densities (percent control) as the percentage of cells treated with vehicle. Data are means ± SEM from three independent experiments. Right panel: Immunoblot of KE7^-^-ETV7-Ex9 cell lysate immunoprecipitated with anti-mTOR or anti-ETV7 antibodies and non-relevant IgG, probed for mTOR and ETV7. (**D**). Top, amino acid sequence of the ETV7 PNT domain with ML and EH sequences highlighted in red and yellow, respectively. Alanine mutations (green) are indicated below the sequence. Bottom left panel: rapamycin response curves of Karpas-299, KE7^-^, KE7^-^-M82A, KE7^-^-I89A, KE7^-^-V105A, and KE7^-^-L109A cells treated with an increasing dose of rapamycin (0.1, 0.3, 1.3, 10, 30, 100, 300, and 1000 ng/mL) for three days or three population doublings. Cell densities (percent control) as the percentage of cells treated with vehicle. Data are means ± SEM from three independent experiments. Bottom right panel: immunoblot of lysates of KE7^-^-M82A, KE7^-^-I89A, KE7^-^-V105A, and KE7^-^-L109A cells immunoprecipitated with anti-mTOR or anti-ETV7 antibodies and nonrelevant IgG, probed for mTOR and ETV7.

**Figure 2 ijms-25-10042-f002:**
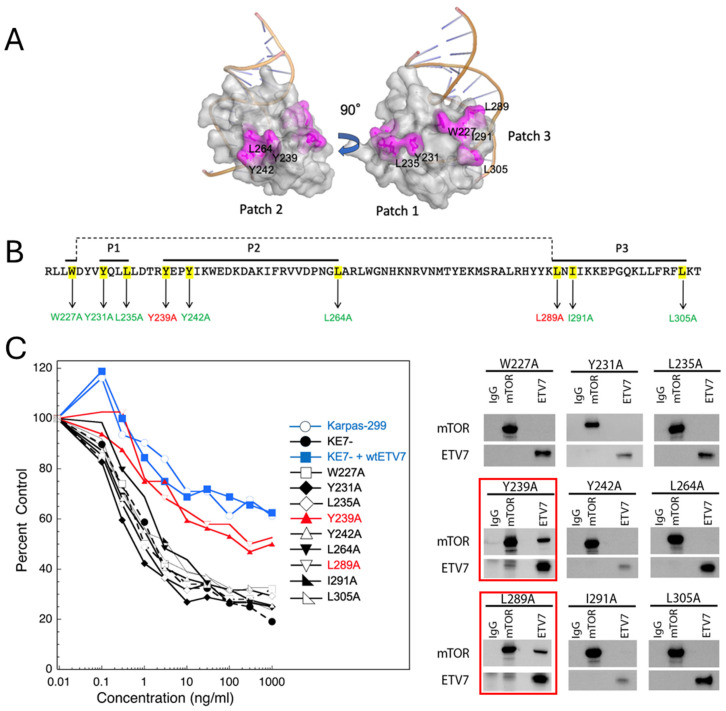
Alanine substitutions in the ETV7 ETS domain identify amino acids involved in the assembly of mTORC3. (**A**). The 3D model derived from the ETV6 ETS domain of the ETV7 ETS domain bound to DNA, showing three hydrophobic patches exposed to solvent (Patch 1, Patch 2, Patch 3) (Y239/Y242/L264; Y231/L235; W227/L289/I291/L305), which can potentially participate in ETS-mediated protein–protein interactions. (**B**). Summary of single alanine substitutions in the ETS domain of ETV7 clustering in three patches (P1, P2, P3) of solvent-exposed hydrophobic amino acids. Cells expressing the ETS mutants indicated in yellow highlighted the amino acid residues mutated, green are sensitive to rapamycin treatment, and those in red are resistant to rapamycin treatment. (**C**). Left panel: rapamycin response curves of Karpas-299, KE7^-^, and KE7^-^-ETV7 cells expressing the ETV7 mutants W227A, Y231A, L235A, Y239A, Y242A, L264A, L289A, I291A, L305A treated with an escalating dose of rapamycin (0.1, 0.3, 1.3, 10, 30, 100, 300, and 1000 ng/mL) for three days or three population doublings. Cell densities (percent control) as the percentage of cells treated with vehicle. Data are means ± SEM from three independent experiments. The SEM is within 5% but cannot be added to the figure due to the density of the plots. Right panel: Immunoblots of lysates of KE7^-^-ETV7^-^-W227A, Y231A, L235A, Y239A, Y242A, L264A, L289A, I291A, L305A cells immunoprecipitated with anti-mTOR or anti-ETV7 antibodies and non-relevant IgG, probed for mTOR and ETV7. The blots boxed in red show the rapamycin-resistant mutants.

**Figure 3 ijms-25-10042-f003:**
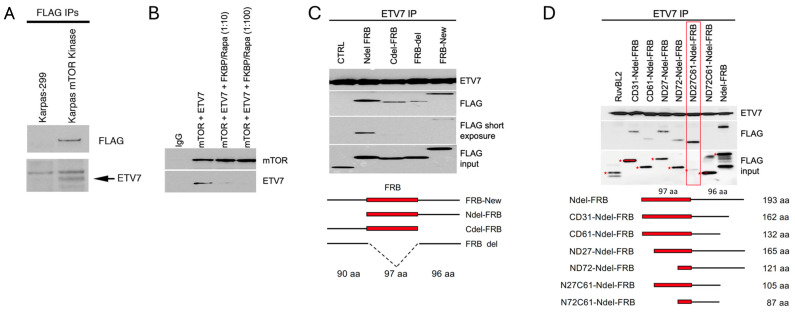
ETV7 binds to the FRB domain of mTOR. (**A**). Immunoblot of the coimmunoprecipitated mTOR kinase domain (amino acids 1363-2549) and ETV7 from Karpas-299 cells expressing the FLAG–mTOR kinase domain with anti-FLAG antibody, probed for FLAG and ETV7. (**B**). Immunoblot of in vitro co-IP of purified ETV7 and mTOR protein in the presence of an increasing amount of FKBP12–rapamycin (mTOR:ETV7:FKBP12/rapamycin = 1:1:0, 1:1:10, and 1:1:100), probed for mTOR and ETV7. (**C**). Immunoblot of ETV7 IP of ETV7 co-incubated with N-terminal and C-terminal deletions of the FRB-new fragment in vitro. Underneath is a schematic representation of serial N-terminal and C-terminal deletions of the FRB-new fragment. Numbers indicate the number of amino acids (aa) in the different segments of the FRB-new fragment. (**D**). FLAG/ETV7 immunoblot of ETV7 IPs of serially deleted Ndel-FRB protein fragments after association with ETV7 in vitro. The lane with the smallest fragment still binding ETV7, N27C61-Ndel-FRB, is boxed in red. The asterisks next to the FRB bands in the input blot indicate the full-length size of each fragment. Underneath is a schematic representation of serial N-terminal and C-terminal deletions of the Ndel-FRB fragment.

**Figure 4 ijms-25-10042-f004:**
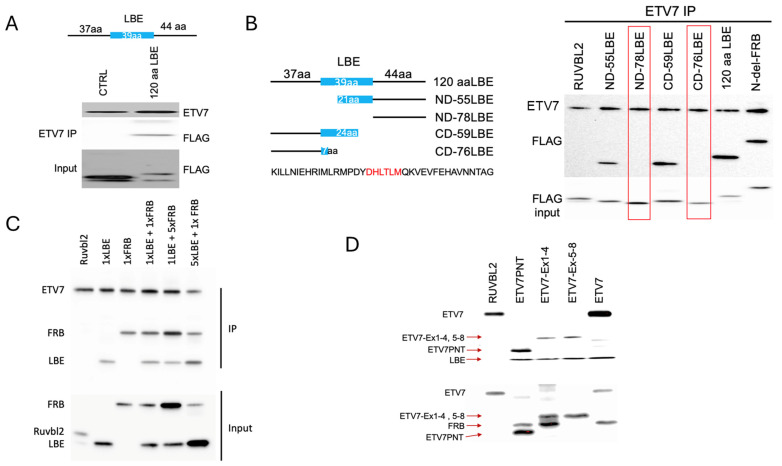
In vitro binding of the LBE sequence of the mTOR kinase domain to ETV7. (**A**). Top drawing: schematic representation of the N- and C-terminally extended LBE (highlighted in blue)fragment. The numbers indicate the number of amino acids in each of the LBE segments. Below is a FLAG immunoblot of an IP of the 120aa LBE fragment after overnight binding to ETV7 in vitro. Control (CTRL) shows the binding of human RUVBL2 to ETV7. (**B**). Left panel, schematic representation of the different deletions of the N- and C-terminal extended LBE fragment (120 amino acids) and underneath the amino acid sequence of the LBE domain with the amino acids essential for ETV7 binding in red; Right, FLAG immunoblot of an ETV7 IP of the different LBE fragments after overnight binding to ETV7 in vitro. Lanes containing fragments that lost binding to ETV7 are boxed in red. (**C**). FRB and LBE bind to ETV7 simultaneously. FLAG immunoblot of an ETV7 IP of LBE (120 amino acids) and Ndel-FRB fragments (ETV7:LBE:Ndel-FRB = 1:1:1, 1:1:5, 1:5:1) after overnight binding to ETV7 in vitro. (**D**). Binding of Ndel-FRB to ETV7 fragments containing the PNT domain and binding of LBE (120 amino acids) to ETV7 fragments containing the PNT and ETS domains in vitro.

**Figure 5 ijms-25-10042-f005:**
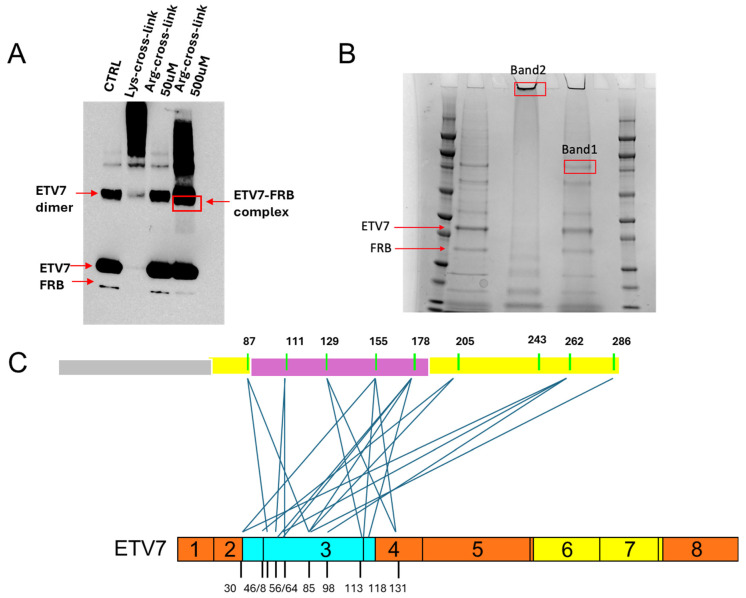
Cross-linking and mass spec identification of ETV7–FRB binding. (**A**). Optimization of cross-linking conditions. From left to right, control showing binding of ETV7 and the FRB-new fragment without cross-linker; binding of ETV7 and FRB-new after treatment with 50 mM lysine crosslinker; ETV7 and FRB-new binding after treatment with 50 mM arginine cross-linker; ETV7 and FRB-new binding after treatment with 500 mM arginine cross-linker. (**B**). Cross-linked samples stained with Coomassie Brilliant Blue. From left to right: marker; control binding of ETV7 and FRB-new without cross-linker; blank; band cut out from the lysine cross-linked material in (**A**); blank (Band 2); band cut out from the material in (**A**) (boxed in red) treated with 500 mM arginine cross-linker (Band 1); blank; marker. (**C**). Diagram of the relative location of cross-linked sites in ETV7 and FRB-new. The central pink color represents the 95 aa FRB domain, while the yellow represents the kinase domain. The FRB-new fragment includes amino acids E^1921^—L^2220^ of mTOR. The blue and yellow color in ETV7 represent the PNT and ETS domains, respectively.

**Figure 6 ijms-25-10042-f006:**
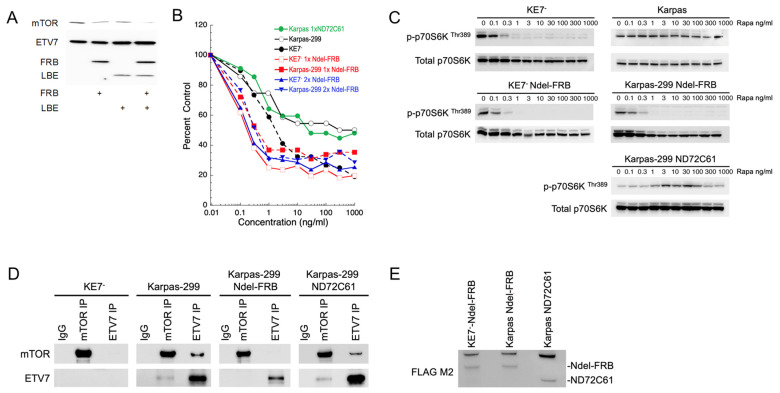
Ndel-FRB competes with ETV7 for mTOR binding and converts Karpas-299 cells from resistant to sensitive to rapamycin. (**A**). Ndel-FRB competes with ETV7 for in vitro binding of mTOR. ETV7 and mTOR were incubated overnight at 4 °C in the presence or absence of Ndel-FRB and/or 120 aa-LBE in vitro. ETV7 Ips were immunoblotted for mTOR, ETV7, and Flag (Ndel-FRB, and 120 aa-LBE). (**B**). Karpas-299 cells that express Ndel-FRB but not ND72C61 become sensitive to rapamycin. Proliferating KE7^-^, Karpas-299, and 2 different Karpas-299 Ndel-FRB cell pools [Karpas transduced once with Ndel-FRB lentivirus (Karpas 1 × Ndel-FRB) and Karpas transduced twice with Ndel-FRB lentivirus (Karpas 2 × Ndel-FRB)] and a Karpas-299 ND72C61 cell pool (Karpas-299 transduced once with ND72C61 lentivirus), as well as 2 different KE7^-^ Ndel-FRB cell pools [KE7^-^ transduced once with Ndel-FRB lentivirus (KE7^-^ 1 × Ndel-FRB) and KE7^-^ transduced twice with Ndel-FRB lentivirus (KE7^-^ 2 × Ndel-FRB)] were treated with increasing amounts of rapamycin for three days or three population doublings. Data are means ± SEM from three independent experiments. The SEM is within 5% but is omitted in the figure due to the density of the plots. (**C**). Cell lysates of Karpas-299, K-E7^-^, Karpas-299–Ndel-FRB, K-E7^-^–Ndel-FRB, and Karpas-299–ND72C61 cells treated with increasing amounts of rapamycin (0, 0.1, 1, 3, 10, 100, 1000 ng/mL), immunoblotted for p-P70S6KThr389, total p70S6K. (**D**). Immunoblots of Co-IPs of ETV7 and mTOR of lysates of Karpas-299, K-E7^-^, Karpas-299–Ndel-FRB, and Karpas-299–ND72C61 cells probed for ETV7, and mTOR. (**E**). Immunoblot showing the presence of Ndel-FRB and ND71C61 in lysates of equal numbers of KE7^-^–Ndel-FRB, Karpas-299–Ndel-FRB, and Karpas-299–ND72C61 cells.

**Table 1 ijms-25-10042-t001:** Mapping of arginine cross-linked sites in the ETV7 PNT domain and the mTOR FRB domain.

ETV7 Position	FRB (mTOR) Position
30	155 (2086)/262 (2193)
46	203 (2134)
48	87 (2018)
56	111 (2042)/178 (2109)
64	111 (2042)/262 (2193)/178 (2109)
85	87 (2019)/286 (2217)/203 (2134)/178 (2109)
98	262 (2193)
113	129 (2060)/155 (2086)
118	178 (2109)
131	129 (2060)/155 (2086)

## Data Availability

The original contributions presented in the study are included in the article/Appendix A, further inquiries can be directed to the corresponding author.

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
