# Peer review of "Assembly of mTORC3 Involves Binding of ETV7 to Two Separate Sequences in the mTOR Kinase Domain"

_ijms, 2024, doi:10.3390/ijms251810042_

Round 1
Reviewer 1 Report
Comments and Suggestions for Authors
Zhan and co-authors presented the manuscript “Assembly of mTORC3 involves binding of ETV7 to two separate sequences in the mTOR kinase domain”. This work is an in-depth study on the characterization of mTORC3, a new complex different from mTORC1 and mTORC2, resistant to rapamycin. In particular, the authors performed a characterization of the specific regions of the mTOR kinase domain able to interact with the transcription factor ETV7.
The work is complete, and does not require further experiments, the data obtained are convincing.
I only have a few minimal observations to suggest to the authors.
- In figure 4B, left panel, is ETV7 the signal present in the upper part of the western blot? It would be useful to indicate what the signals in the western blot represent.
- In the caption of figure 4A, the control of RUVBL2 binding to ETV7 is reported, which is probably referred to figure 4C and 4D. It would be useful to make a reference to this control also in the text.
- The sentence “showing that these alanine substitutions in both the ML and EH interfaces contribute to the assembly of mTORC3” is confusing, please rephrase.
- Figure 5 seems to be low resolution. In particular, figure 5C, the amino acid letters are not readable.
- In the legend of figure 5C, it would be useful to report the description of what the segments with the various colors represent.
Author Response
Zhan and co-authors presented the manuscript “Assembly of mTORC3 involves binding of ETV7 to two separate sequences in the mTOR kinase domain”. This work is an in-depth study on the characterization of mTORC3, a new complex different from mTORC1 and mTORC2, resistant to rapamycin. In particular, the authors performed a characterization of the specific regions of the mTOR kinase domain able to interact with the transcription factor ETV7.
The work is complete, and does not require further experiments, the data obtained are convincing.
I only have a few minimal observations to suggest to the authors.
- In figure 4B, left panel, is ETV7 the signal present in the upper part of the western blot? It would be useful to indicate what the signals in the western blot represent.
Response: We thank the reviewer for the suggestion and altered Fig. 4B accordingly.
- In the caption of figure 4A, the control of RUVBL2 binding to ETV7 is reported, which is probably referred to figure 4C and 4D. It would be useful to make a reference to this control also in the text.
Response: We included this suggestion in the text in red.
- The sentence “showing that these alanine substitutions in both the ML and EH interfaces contribute to the assembly of mTORC3” is confusing, please rephrase.
Response: We changed the sentence as follows, please see the text in red:
All four mutants were sensitive to rapamycin, thereby precluding mTORC3 assembly, indicating that these amino acid residues in both the ML and EH surfaces contribute to the assembly of mTORC3.
- Figure 5 seems to be low resolution. In particular, figure 5C, the amino acid letters are not readable.
Response: We fully agree with the reviewer, but due to space restriction and the density of the lettering in Fig. 5 we are unable to expand the size of the FRB fragment so that the amino acid sequence becomes legible. The next best solution we could come up with is that we mention the amino acid positions of the fragment E1921 – L2220 in the legend of the figure 5C. As the amino acid lettering is not legible in figure 5c, we made a new figure omitting the amino acid sequence. A part of the FRB amino acid sequence is available in supplementary Fig 3.
To fully clarify Fig. 5C, we added the following information to the legend:
The central pink color represents the 95 aa FRB domain, while the yellow represents the kinase domain. The FRB-new fragment includes amino acids E1921 – L2220 of mTOR. The blue and yellow color in ETV7 represent the PNT and ETS domains, respectively.
Reviewer 2 Report
Comments and Suggestions for Authors
This research article was detail to assay the key domains and residues for ETV7 and mTORC3 interaction. Because ETV7 is only one known key molecule to assemble mTORC3 to control rapamycin sensitivity, to identify the small molecule to block ETV7- mTORC3 interaction is the big issue for anticancer propose in ETV7 highly expression cancers. Overall assay is no big problem, but there are several questions needed to address.
1. Why Karpas-299-Ndel-FRB cells can much more sensitive to rapamycin than the non EVT7 expressed cells (K-E7-) in ~ 1 ng/c.c rapamycin treatment (Fig 6B)?
2. Why Fig 6G has a big and clear size of blotting in anti-Flag staining in all three expression cells? It seems a non-specific band but much clear than the specific band as authors' labeling.
3. Ndel-FRB fragment seems like a powerful mTORC3 inhibitor for anti-cancer propose, but the assay is still too preliminary and limited in this study. It’s needed to assay more than one cancer cell line with highly ETV7/mTORC3 expressed to evaluate anti-cancer activity under low concentration of rapamycin treatment.
Author Response
This research article was detail to assay the key domains and residues for ETV7 and mTORC3 interaction. Because ETV7 is only one known key molecule to assemble mTORC3 to control rapamycin sensitivity, to identify the small molecule to block ETV7- mTORC3 interaction is the big issue for anticancer propose in ETV7 highly expression cancers. Overall assay is no big problem, but there are several questions needed to address.
- Why Karpas-299-Ndel-FRB cells can much more sensitive to rapamycin than the non EVT7 expressed cells (K-E7-) in ~ 1 ng/c.c rapamycin treatment (Fig 6B)?
Response: This is a great question, we do not fully understand the mechanism, but we believe that the FRB fragment may interfere with mTORC1 activity, which is somewhat supported by the p70S6 phosphorylation blots (Fig 6C). When one carefully looks at the p70S6 phosphorylation blot of KE7- cells, there is still some phosphorylation even at high concentrations of rapamycin, while phosphorylation is absent in the Ndel-FRB KE7- and Ndel-FRB Karpas blots.
To support this possibility, we checked the available literature. Using the mTOR structure information, other investigators found that FRB acts as a gatekeeper by restricting access to the active site, while also granting privileged substrate access through its binding site for the secondary motif (reference 12), therefore the FRB domain plays an additional role in controlling mTORC1 activity.
The authors in reference 12 tested whether the isolated FRB domain inhibits phosphorylation in trans. Their Figure 4c shows that wildtype FRB but not a Ser2035Ile FRB mutant inhibits overall p70S6K1 phosphorylation by 50% and phosphorylation of Thr 389 by 75% at the highest concentration tested.
Therefore, based on our observation and that of others, the expression of FRB alone apparently interferes with the activity of mTOR. We believe that this provides a reasonable explanation as to why Karpas-299-Ndel-FRB and KE7- -Ndel-FRB cells are more sensitive to rapamycin than KE7 cells alone at rapamycin concentrations lower than 3 ng/ml (Fig 6B).
As a note of clarification to the reviewer (not as part of this paper) we wish to point out that mTORC3 assembly does not depend on high levels of ETV7 expression. Even low levels of ETV7 (as in Karpas-299, which we estimate to contain 2 atto gram ETV7/cell or 12000 molecules/cell) are sufficient to accomplish this. In fact, the amount of mTORC3 that cells can accommodate is restrictive and increasing the level of ETV7 expression does not boost the amount of mTORC3 beyond this restrictive level.
- Why Fig 6G has a big and clear size of blotting in anti-Flag staining in all three expression cells? It seems a non-specific band but much clear than the specific band as authors' labeling.
Response: The prominent band is a background band of the FLAG antibody in Fig. 6E. We are sure that the other bands represent the two different Ndel-FRB fragments because they are of the predicted size and the difference in size between the two is exactly what would be predicted based on their amino acid sequence.
- Ndel-FRB fragment seems like a powerful mTORC3 inhibitor for anti-cancer propose, but the assay is still too preliminary and limited in this study. It’s needed to assay more than one cancer cell line with highly ETV7/mTORC3 expressed to evaluate anti-cancer activity under low concentration of rapamycin treatment.
Response: We fully agree with the suggestion of the reviewer. We would love to test more cancer cell lines and cultured primary tumor cells that express ETV7/mTORC3 to evaluate the anti-cancer activity of the Ndel-FRB fragment during treatment with low concentrations of rapamycin. Unfortunately, we are unable to do that due to the retirement of Dr. Grosveld and the closure of the lab. We now state in the discussion “In addition, we only showed the inhibitory activity of Ndel-FRB in Karpas-299 cells, but further experiments need to be performed in other cell lines and primary tumor cells to determine whether our observations are applicable to mTORC3 expressing cells in general. We hope that our results will inspire other investigators to pursue this line of investigation.
Round 2
Reviewer 2 Report
Comments and Suggestions for Authors
Nowadays, no animal model or real clinical sample articles were not accepted by IF>5 (2023IF) or >4 (2024IF). This is my experiences for over than 100 times of inviting reviewer works. This is why we add the new animal experiment for IF>4 or directly submit to IF~3 in this year when there is no clinical sample or animal model in the article. Because mTORC3 is still novel molecule in cancer research fiedl, so only containing in vitro assay seems like still acceptable. But only one cell line assay can not be allowed for publication for IF>4, especially the new molecule for anticancer purpose.